# Evaluation of Nitric Oxide-Donating Properties of 11*H*-indeno[1,2-*b*]quinoxalin-11-one Oxime (IQ-1) by Electron Paramagnetic Resonance Spectroscopy

**DOI:** 10.3390/molecules29163820

**Published:** 2024-08-12

**Authors:** Viacheslav V. Andrianov, Igor A. Schepetkin, Leah V. Bazan, Khalil L. Gainutdinov, Anastasia R. Kovrizhina, Dmitriy N. Atochin, Andrei I. Khlebnikov

**Affiliations:** 1Zavoisky Physical-Technical Institute of the Russian Academy of Sciences, Kazan 420029, Russia; andrianov.neuro@yandex.ru (V.V.A.); l_v_bazan@mail.ru (L.V.B.); kh_gainutdinov@mail.ru (K.L.G.); 2Department of Human and Animal Physiology, Institute of Fundamental Medicine and Biology, Kazan Federal University, Kazan 420008, Russia; 3Kizhner Research Center, Tomsk Polytechnic University, Tomsk 634050, Russia; igor@montana.edu (I.A.S.); anaskowry@gmail.com (A.R.K.); 4Department of Microbiology and Cell Biology, Montana State University, Bozeman, MT 59717, USA; 5Cardiovascular Research Center, Massachusetts General Hospital, Harvard Medical School, Charlestown, MA 02115, USA; datochin@mgh.harvard.edu

**Keywords:** c-Jun N-terminal kinase (JNK) inhibitor, IQ-1 (11*H*-indeno[1,2-*b*]quinoxalin-11-one oxime), electron paramagnetic resonance, nitric oxide donor, spin trap, hemoglobin, DFT calculation

## Abstract

IQ-1 (11*H*-indeno[1,2-*b*]quinoxalin-11-one oxime) is a specific c-Jun N-terminal kinase (JNK) inhibitor with anticancer and neuro- and cardioprotective properties. Because aryloxime derivatives undergo cytochrome P450-catalyzed oxidation to nitric oxide (NO) and ketones in liver microsomes, NO formation may be an additional mechanism of IQ-1 pharmacological action. In the present study, electron paramagnetic resonance (EPR) of the Fe^2+^ complex with diethyldithiocarbamate (DETC) as a spin trap and hemoglobin (Hb) was used to detect NO formation from IQ-1 in the liver and blood of rats, respectively, after IQ-1 intraperitoneal administration (50 mg/kg). Introducing the spin trap and IQ-1 led to signal characteristics of the complex (DETC)_2_-Fe^2+^-NO in rat liver. Similarly, the introduction of the spin trap components and IQ-1 resulted in an increase in the Hb-NO signal for both the R- and the T-conformers in blood samples. The density functional theory (DFT) calculations were in accordance with the experimental data and indicated that the NO formation of IQ-1 through the action of superoxide anion radical is thermodynamically favorable. We conclude that the administration of IQ-1 releases NO during its oxidoreductive bioconversion *in vivo*.

## 1. Introduction

Ischemia–reperfusion can result in organ failure through a complicated series of events due to intracellular injury. These events include an activation of *c*-Jun N-terminal kinase (JNK) [1] and a reduction in nitric oxide (NO) production [2]. Furthermore, investigating the pathophysiology of brain injury indicates the potential role of NO as a protective molecule [3]. In the brain, the effects of NO on ischemic injury are thought to be dependent on the sources of its production and the stage of the ischemic process [4,5]. NO is produced by endothelial, neuronal, and inducible NO synthases (eNOS, nNOS, and iNOS) [6]. eNOS and nNOS are constitutively expressed in endothelial cells and neurons, respectively, and the expression of iNOS is prevalent in macrophages [7]. The low concentration of NO that is produced by eNOS confers protective effects during cerebral ischemia [4,8]. Based on the coupling of NO and JNK pathways [9] and the protective role of exogenous NO in reperfusion-induced brain injury [5,8], we hypothesized that agents with dual functions as JNK inhibitors and NO donors could have neuroprotective effects against cerebral ischemic and reperfusion injury. To date, oxime derivatives have been demonstrated *in vivo* and *ex vivo* as NO donors [10,11]. The biotransformation of oximes on microsomal cytochrome P450 (CYP450) results in NO and keto derivatives [12,13].

Previously, we described a specific JNK inhibitor **IQ-1**, 11*H*-indeno[1,2*-b*]quinoxalin-11-one oxime [14,15]. In the models of global and focal cerebral ischemia, the compound **IQ-1** effectively protected against stroke injury [16,17,18]. It was also found that a major metabolite of **IQ-1** is ketone **IQ-18** [19]. Taking into account that the neutral form of **IQ-1** has an oxime group, we suggested that, similarly to other aryloxime derivatives [10,12,13], this compound could release NO during its redox bioconversion (Figure 1):

One of the most effective methods for detecting NO in biological tissues has become the electron paramagnetic resonance (EPR) approach using spin-trapping [20,21,22,23,24]. The spin trap method is based on the reaction of a radical (in this case, NO) with an iron-containing spin trap: the trapping complex is diamagnetic, and its central Fe^2+^ ion has a high affinity for NO radicals [25]. As a result of the reaction, an adduct with a characteristic EPR spectrum is formed. In our experiments, the Fe^2+^ complex with diethyldithiocarbamate (DETC) was used to capture NO and form a stable ternary complex (DETC)_2_–Fe^2+^–NO in various animal tissues. This complex is characterized by an easily recognizable EPR spectrum with g-factor in the range 2.035–2.040 and a triplet hyperfine structure [21,22,26]. The method has a sensitivity of 0.04–0.4 nM [22]. In addition, hemoglobin (Hb) inside red blood cells forms a complex with NO exhibiting a specific EPR signal which could be usable for NO registration in tissues [27,28]. Thus, we used the EPR-based approach to register NO production in liver and blood of rats after **IQ-1** administration. We found an increase in the (DETC)_2_–Fe^2+^–NO signal levels in the liver and HbFe^2+^–NO signals for both the R- and the T-conformers in blood with **IQ-1** in comparison with the administration of the trap alone. The density functional theory (DFT) calculations were performed to substantiate the proposed mechanism of **IQ-1** oxidation by the CYP450-originated superoxide anion radical (O_2_**·**^−^). The processes of NO trapping by (DETC)_2_–Fe^2+^ complexes were also studied by the DFT method.

## 2. Results and Discussion

### 2.1. EPR Signals of Samples from Liver

No NO-related EPR signals were registered in the control (group I, no **IQ-1** and the spin trap) and the group where **IQ-1** was administered (group II) in samples from the liver. The application of the spin trap (group III) led to the appearance of a NO-related EPR signal, associated with the natural production of NO. The introduction of both **IQ-1** and the spin trap (group IV) led to signal characteristics of the complex (DETC)_2_–Fe^2+^–NO (Figure 1), confirming the formation of NO in the organism. However, the signal amplitude for samples with **IQ-1** in comparison with the trap alone was only ~2-fold higher. This could be explained by enhanced O_2_**·**^−^ formation on the CYP450 oxidoreductase [29], which led to the degradation of the (DETC)_2_–Fe^2+^–NO complex and the decrease in the EPR signal [30]. It should be noted that the spin trap was used by us for the first time to record NO production during the bioconversion of aryloximes *in vivo*. Previously, the incubation of *para*-hexyloxybenzamidoxime (**AL1**) and 4-(chlorophenyl)methyl ketone oxime with microsomes in the presence of NADPH resulted in the appearance of characteristic EPR signals from the complex P420/P450–Fe^2+^–NO [31]. Future studies are needed to optimize the method and increase efficiency of NO trapping using various doses of **IQ-1** and the spin trap, as well as different time points between spin trap/**IQ-1** injections and tissue sampling.

### 2.2. EPR Signal of Blood Samples

Blood samples were received from the same animals used in the experiment with a spin trap. A low EPR signal level in the blood samples of the control (group I) was registered. The signal is associated with the natural production of NO in the blood vessels by eNOS and the formation of HbFe^2+^–NO complexes. In group II (**IQ-1** without the spin trap), a low signal level is also observed in the form of the R-conformer and the T-conformer of the HbFe^2+^–NO complexes. In group III (the spin trap alone), there are small signals of the R- and T-conformers, similar to the control. In group IV (both **IQ-1** and the spin trap were injected), the introduction of spin trap components led to an increase in the interaction of NO with HbFe^2+^, and large signals from both the R- and the T-conformers were observed (Figure 2). The signal levels of R- and T-conformers with **IQ-1** were ~4–6 times higher compared to those of the control and samples with the trap alone.

### 2.3. A DFT Study of NO Complexes and NO Formation from **IQ-1**

In accordance with the obtained data, the injection of **IQ-1** without the spin trap (group II) did not lead to a significant increase in HbFe^2+^–NO signals, in contrast to group IV (see above). This can be explained by the appearance of free Fe^2+^ ions in the blood of group IV animals (as one of the injected parts of the spin trap), which can serve as NO transporters. Indeed, the dissociation constants of complexes formed with NO by different non-heme Fe^2+^ species, including hydrated iron [Fe(H_2_O)_6_]^2+^, are within 10^−2^–10^−6^ M, while binding NO to HbFe^2+^ in R- and T-states is characterized by the dissociation constants of about 10^−12^ and 10^−10^ M, respectively [32]. Hence, NO can be captured by HbFe^2+^ from the Fe^2+^–NO species, resulting in the increase in HbFe^2+^–NO signals in the experimental EPR spectra. On the other hand, NO cannot be transferred to hemoglobin from its complex formed with the spin trap. Thus, the Gibbs free energy ΔG°_298_ of the dissociation process (1) estimated by the DFT method is equal to +25.0 kcal/mol, indicating very tight NO binding with the (DETC)_2_–Fe^2+^ spin trap.
(DETC)_2_–Fe^2+^–NO → (DETC)_2_–Fe^2+^ + NO(1)

The optimized geometries of the ferrous complexes participating in the reaction are shown in Figure 3. The isotropic g-factor calculated by the DFT method for (DETC)_2_–Fe^2+^–NO complex is equal to 2.045, which is close to the experimental data [21,22,26].

We also considered the possibility of binding one or two water molecules by the (DETC)_2_–Fe^2+^ complex. However, these processes are non-spontaneous, as they are accompanied by positive changes in Gibbs free energy of +3.1 and +11.2 kcal/mol in aqueous solution for binding the first and the second water molecules, respectively. With one water molecule as a co-ligand to NO in the hypothesized H_2_O–(DETC)_2_–Fe^2+^–NO complex, we did not obtain a stable structure for this coordination compound due to a cleavage of the Fe···OH_2_ bond during the DFT geometry optimization. This result is in agreement with the known ability of NO to expel *trans*-ligands from the coordination sphere of a ferrous complex [32]. However, other ligands with higher coordination capabilities can promote the dissociation of the trapped NO.

According to the literature data, ketoximes and amidoximes can be oxidized by CYP450 with the release of NO via the intermediate formation of O_2_**·**^−^, the latter being a dissociation product of the CYP450–Fe^2+^–O_2_ complex [31]. The proposed mechanism of the C=N−OH group oxidative transformation “outside the active site” [33] involves the nucleophilic attack of O_2_**·**^−^ to the oxime carbon atom and the further transfer of the electron pairs within the anion–radical adduct A**·**^−^ (Figure 2).

To investigate the possibility of the oxime group oxidation by O_2_**·**^−^, we calculated Gibbs free energies (ΔG°_298_) of steps (1) and (2) (Figure 2) using the DFT method for the conversion of **IQ-1** into NO and the corresponding ketone, in comparison with amidoxime substrate **AL1** (Table 1).

Compound **AL1** is a well-known NO donor [12] with experimental support for the NO formation “outside the active site” [31]. The optimized geometric structures of adduct A**·**^−^ obtained from compounds **IQ-1** and **AL1** are shown in Figure 4. In these adducts, the oxime hydrogen atom is shifted towards the attached dioxygen fragment. Thus, the OO···H interatomic distances equal 0.998 and 1.023 Å for **IQ-1**- and **AL1**-derived adducts, respectively, while the corresponding lengths of NO···H bonds are enhanced up to 1.684 and 1.557 Å. The O−O bond in the adducts is lengthened by 0.11–0.12 Å as compared to the optimized O−O distance in O_2_**·**^−^ (1.346 Å). On the other hand, the N−O interatomic distance in the adducts is shortened by 0.077 and 0.096 Å with respect to the optimized N−O bond lengths in the oxime and amidoxime substrates **IQ-1** and **AL1** (1.365 and 1.427 Å, respectively). These geometric changes are in accordance with the subsequent formation of OH^−^ and NO species on step (2) (Figure 2).

It should be noted that the starting compound **AL1** is thermodynamically more stable in the form of *Z*-isomer (ΔG°_298_ of *E*→*Z* isomerization equals −5.36 and −4.97 kcal/mol for the gas phase and aqueous solution, respectively). However, for **IQ-1,** the *Z*- and *E*-configurations of the oxime moiety are more favorable in the gas phase and aqueous medium (ΔG°_298_ of *E*→*Z* isomerization are −1.30 and +0.61 kcal/mol, respectively). Previously, a higher thermodynamic stability of **IQ-1** *E*-isomer in other solvents was reported [19,34]. For each starting oxime-containing compound, the isomer with a lower (or more negative) Gibbs free energy was used in the calculation of the values shown in Table 1.

In the gas phase, both steps (1) and (2) are characterized by negative ΔG°_298_ values, resulting in ΔG°_298_ = −37.91 and ΔG°_298_ = −50.15 kcal/mol for the whole oxidation process described by Figure 2 for compounds **IQ-1** and **AL1** [see the “(1) + (2)” entries in Table 1]. It is noteworthy that the change in Gibbs energy on the formation of adduct A**·**^−^ on step (1) is more negative for **IQ-1** than for the known NO donor **AL1**. We also applied the approximate CPCM solvation model to account for the solvent effects on the reaction thermodynamics. This approximation resulted in positive ΔG°_298_ values for the formation of anion–radical adduct A**·**^−^, mainly due to a very high calculated solvation energy of the O_2_**·**^−^ participating in step (1). Nevertheless, the conversion of A**·**^−^ into the final products in step (2) in the aqueous solution, according to the DFT data, is accompanied by a substantial decrease in Gibbs free energy both for **IQ-1** and **AL1**, indicating the spontaneous character of the whole oxidation process (Table 1). The more negative ΔG°_298_ values obtained with the account of water solvation can be explained by the high calculated CPCM solvation energy of the hydroxide anion as the reaction product.

## 3. Materials and Methods

### 3.1. Animals

This research was performed in accordance with the EU Directive 2010/63/EU concerning the protection of animals used for scientific purposes and was approved by the Animal Care and Use Committee of the Kazan Federal University. Experiments were performed on 20 adult male Wistar rats (weight: 250–280 g). Rats were housed in groups of five animals per cage (57 × 36 × 20 cm) under standard laboratory conditions (ambient temperature of 22 ± 2 °C, relative humidity of 60%, and 12:12 h light–dark cycle) in cages with sawdust bedding and were provided standard rodent feed and *ad libitum* water access.

### 3.2. Chemicals and the Studied Compound

Iron sulfate (FeSO_4_ · 7H_2_O), sodium citrate (C_6_H_5_Na_3_O_7_ · 2H_2_O), and DETC sodium salt trihydrate (C_5_H_10_NNaS_2_ · 3H_2_O) were purchased from Sigma (USA). **IQ-1** was synthesized as described previously [35]. The **IQ-1** chemical structure was confirmed by mass spectrometry and nuclear magnetic resonance. The purity of the sample was 99.9%.

### 3.3. Experimental Protocol

Sodium DETC was introduced intraperitoneally (*i.p.*) in the concentration of 500 mg/kg in 2.5 mL water [36]. The solution mixture, comprising 37.5 mg/kg iron sulfate and 187.5 mg/kg sodium citrate (all in 1 mL water to one animal) and prepared before injection, was injected under the skin at three locations—left and right legs and withers. In aqueous solution, iron sulfate and sodium citrate produce iron citrate. Sodium DETC and iron citrate are distributed in an organism and their interactions generate the water-insoluble hydrophobic (DETC)_2_-Fe^2+^ complex (spin trap), which can interact with NO to form the paramagnetic mononitrosyl iron complex (MNIC) (DETC)_2_-Fe^2+^–NO detectable by EPR spectroscopy [23].

Animals were randomly divided into 4 groups, with 5 rats in each: group I (control), underwent only tissue sampling without the introduction of chemicals; group II (**IQ-1**), **IQ-1** (50 mg/kg) was injected *i.p*. 60 min before tissue sampling, no spin trap was used; group III (spin trap), components of the spin trap were injected, then after 60 min, tissue sampling was performed; group IV (spin trap + **IQ-1**), components of the spin trap were injected, then **IQ-1** (50 mg/kg) was injected *i.p.* after 20 min, and tissues were collected after 40 min.

The measurements were taken on an EPR spectrometer Bruker EMX/plus (Billerica, MA, USA) with a temperature module ER 4112HV at X-band with the following parameters: modulation, 100 kHz; amplitude modulation, 2 G; microwave power, 30 mW; time constant, 200 ms; and temperature, 77 K [37]. All experiments were performed in a Dewar flask without microwave saturation and overmodulation. Microwave parameters were identical for all the samples [38].

The liver and blood samples collected after the injections were weighed before the EPR experiments. All the measurements were performed on two paramagnetic complexes of the Fe^2+^ ion with NO, including the MNIC based on the spin trap and the iron hemoglobin complex with NO (HbFe^2+^–NO). Blood EPR spectra represent a sum of the individual EPR signals of 6- (R-conformer) and 5-coordinate (T-conformer) nitrosyl hemes [39,40,41]. The amplitude of the EPR spectra was normalized to the sample weight and amplitude of the EPR signal of the reference samples with known concentrations, as described previously [26,42].

### 3.4. DFT Calculations

The ORCA 5.0.4 computational chemistry software [43] was used for the DFT calculations of the starting compounds and intermediate and final products of the oxime-containing substrate oxidation by O_2_**·**^−^. Geometry optimizations were carried out through the unrestricted Kohn–Sham method using the B3LYP/G functional [44] with the 6-311+G(d,p) basis set. The compounds participating in NO binding by the (DETC)_2_-Fe^2+^ spin trap were calculated with the PBE0 functional [45] and def2-TZVPD basis set [46,47]. The D3BJ dispersion correction [48] was used in all the calculations. To account for solvent effects, the conductor-like polarizable continuum solvation model (CPCM) [49] was applied with water as a solvent. The normal vibration analysis was performed for the optimized geometries to confirm the attainment of the energy minima and calculate the thermochemistry data. The analysis and visualization of the DFT results were carried out with the Chemcraft 1.8 program. The ORCA 5.0.4 output files containing full DFT results for some of the investigated compounds can be found in the Appendix A.

The DFT functionals used in our study were chosen based on the literature data. Thus, PBE0 is one of the functionals recommended for the calculations of transition metal complexes [50], while B3LYP (or B3LYP/G in the ORCA software notation) is conventionally used for the investigation of organic reactions and intermediates [51]. The “triple-zeta” quality basis sets 6-311+G(d,p) and def2-TZVPD containing diffuse functions are suitable for most chemical applications [52].

The isotropic g-factor of the (DETC)_2_–Fe^2+^–NO complex was calculated using the long-range corrected LC-BLYP functional [53] and the correlation consistent aug-cc-pVTZ basis set [54] with the molecular geometry of the complex optimized as described above.

## 4. Conclusions

Thus, using the EPR method with the spin trap, EPR spectra specific to the (DETC)_2_–Fe^2+^–NO and HbFe^2+^–NO complexes in rat liver and blood samples after **IQ-1** administration were obtained. The obtained results indicate that NO is formed *in vivo* through the oxidation of **IQ-1**. The additional DFT calculations show the possibility of NO formation via the interaction of CYP450-derived O_2_**·**^−^ with **IQ-1**, which is in agreement with our EPR experiments. The dual function of **IQ-1** as a JNK inhibitor and NO donor may contribute to the neuroprotective, anticancer, and other pharmacological effects of the compound.

## Data Availability

Data are contained within the article and the Appendix A.

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
