# Peer review of "Evaluation of Nitric Oxide-Donating Properties of 11H-indeno[1,2-b]quinoxalin-11-one Oxime (IQ-1) by Electron Paramagnetic Resonance Spectroscopy"

_molecules, 2024, doi:10.3390/molecules29163820_

Round 1

Reviewer 1 Report

Comments and Suggestions for Authors

 Dear colleagues,

I found very interesting your investigation. As I understood you tried to detect the NO releasing from IQ-1 in rat liver induced by cytochrome P450. Very NO effective spin trap (DETC+Fe2+) wass used in your experiments. You demonstrated that really the NO trap functioned in the liver. However the fact remained without explanation why used NO trap could nod catch high NO amount which could pass over from liver to blood?

Why the authors observed large EPR signal from HB-NO in blood despite the NO trap (DETC+Fe2+) addition to the rats? It means that the trap could not catch NO released from IQ-1 in liver. By the way, the authors failed to detect large EPR signal from Hb-NO when IQ-1 was added to the animals alone without NO spin trap.

Author Response

Comment 1: However, the fact remained without explanation why used NO trap could not catch high NO amount which could pass over from liver to blood? 

Response 1: This is the first study to evaluate NO production by aryloximes using the DETC-Fe2+ spin trap in vivo. Indeed, our study showed that the spin trap does not capture large amounts of NO in the liver that pass from the liver into the blood. Unlike sodium nitroprusside and other known NO donors that can produce NO spontaneously under the appropriate reducing conditions, aryloximes generate NO via enzymatic bioconversion. Many factors can affect the dynamics and efficiency of NO enzyme production in the liver. We believe that the method could be optimized using different doses of IQ-1 and spin trap, as well as various time points between trap and IQ-1 injections and tissue sampling to increase the efficiency of NO trapping. This is the limitation of this study, and a comprehensive investigation on this topic in our further experiments. This limitation was added in the discussion (see lines 91-98). It should be noted that the spin trap is insoluble and not easily cleared in parenchymatous organs (e.g., the liver). During the bioconversion of aryloximes, superoxide anion can be produced by cytochrome P450, and this free oxygen radical can destroy the NO complex DETC-Fe2+-NO in liver (https://doi.org/10.2741/3538) (lines 87-90). On the other hand, we cannot exclude that the bioconversion of IQ-1 with the formation of NO may occur in the blood via another CYP450-independent enzymatic pathway, for example, through bioconversion by NO synthases. Indeed, Glover et al. (1999) didn’t find a characteristic EPR signal in the liver after administration of cyclohexanone oxime (200 mg/kg), although they found EPR signals of nitrosyl complexes in the blood (https://doi.org/10.1021/tx990058v).

Comment 2: Why the authors observed large EPR signal from HB-NO in blood despite the NO trap (DETC+Fe2+) addition to the rats? It means that the trap could not catch NO released from IQ-1 in liver. By the way, the authors failed to detect large EPR signal from Hb-NO when IQ-1 was added to the animals alone without NO spin trap.

Response 2: 

We hypothesize that Fe2+ ions (introduced as part of the EPR trap) may scavenge NO and transport it from the liver to the blood for subsequent binding to hemoglobin. Our DFT calculations and literature data [Cooper, C.E., Nitric oxide and iron proteins. Biochimica Et Biophysica Acta-Bioenergetics, 1999. 1411, 290-309] showed that NO can be captured by HbFe2+ from the Fe2+–NO species, giving the increase of HbFe2+–NO signals in the experimental EPR spectra (lines 130-133).  This mechanism may explain why we did not detect a large EPR signal from Hb-NO when IQ-1 was added to the animals alone without a NO spin trap.

Reviewer 2 Report

Comments and Suggestions for Authors

Andiranov and coworkers have prepared a manuscript on utilizing 11H-indeno[1,2-b]quinoxalin-11-one oxime (IQ-1) as a nitric oxide (NO) donor in biological systems. The NO delivery results were tested using EPR spectroscopy, with (DETC)2-Fe2+ used as a spin trap to track NO generation. DFT calculations were performed to reveal the interactions between (DETC)2-Fe2+ and NO. Additionally, the reaction energetics of the activities of IQ-1 and AL1 with superoxide (O2.-) to generate NO were also explored using DFT calculations. Both experimental and DFT calculation results showed that IQ-1 could be a promising candidate as an NO donor. The manuscript was well-prepared and could be published if the following questions/comments are addressed:

  1. The authors used EPR to monitor NO generation. From the results shown in Figures 1 and 2, the nature of the spins is anisotropic, making it inaccurate to describe the resonance with merely isotropic g values. The spectra exhibit multiple peaks and splitting, suggesting that they cannot be described using only one giso value. Since the EPR spectra were fitted, please provide the anisotropic g values (or g tensor, i.e., g1, g2, g3) from the spectra.
  2. As NO is a non-innocent ligand, what is the oxidation state of the Fe and NO in the (DETC)2-Fe2+-NO system? It is known that electron transfer between Fe and NO is possible. Is it accurate to describe it as (DETC)2-Fe2+-NO, or should it be (DETC)2-Fe3+-NO-? Any evidence from experiments and/or DFT calculations?
  3. The authors used the calculated ΔG to evaluate the tight NO binding of (DETC)2-Fe2+. However, there could be solvent-assisted dissociation of NO involving the addition of an H2O molecule in the aqueous system to form H2O-(DETC)2-Fe2+-NO and then release the NO. This pathway might have a lower ΔG. Please evaluate this pathway as well.
  4. Since DFT calculations were performed, what are the g values obtained from DFT, and how do they compare to the experimental ones?

Author Response

Comment 1:  The authors used EPR to monitor NO generation. From the results shown in Figures 1 and 2, the nature of the spins is anisotropic, making it inaccurate to describe the resonance with merely isotropic g values. The spectra exhibit multiple peaks and splitting, suggesting that they cannot be described using only one giso value. Since the EPR spectra were fitted, please provide the anisotropic g values (or g tensor, i.e., g1, g2, g3) from the spectra. 

Response 1: The g-tensor has axial symmetry [Vanin, A.F. et al., Iron dithiocarbamate as spin trap for nitric oxide detection: Pitfalls and successes. Nitric Oxide, Pt D, 2002, 359, 27-42]. The signals in the EPR spectrum corresponding to the perpendicular component of the g-tensor are well expressed in the spectrum, so we consider only the perpendicular component of the g-tensor. The signals corresponding to the parallel component of the g-tensor are less intense and the signal-to-noise ratio does not allow us to accurately determine their intensities. Since the accuracy of determining the signal intensity is important to us, our results are based on a comparison of the perpendicular component intensity of the g-tensor. The (DETC)2-Fe2+-NO complex has an anisotropic g-tensor: g perpendicular = 2.038 and g parallel = 2.02.

Comment 2: As NO is a non-innocent ligand, what is the oxidation state of the Fe and NO in the (DETC)2-Fe2+-NO system? It is known that electron transfer between Fe and NO is possible. Is it accurate to describe it as (DETC)2-Fe2+-NO, or should it be (DETC)2-Fe3+-NO-? Any evidence from experiments and/or DFT calculations?

Response 2:  According to our DFT results, the electric charge of the NO ligand in (DETC)2-Fe2+-NO complex is equal to -0.270 or -0.176 in Mulliken or Hirshfeld approximations, respectively. Thus, the electron transfer from the spin trap to the NO molecule is not very significant. Therefore, we consider that the oxidation state of Fe was not changed on the complex formation.

Comment 3: The authors used the calculated ΔG to evaluate the tight NO binding of (DETC)2-Fe2+. However, there could be solvent-assisted dissociation of NO involving the addition of an H2O molecule in the aqueous system to form H2O-(DETC)2-Fe2+-NO and then release the NO. This pathway might have a lower ΔG. Please evaluate this pathway as well. 

Response 3: We have found that one or two water molecules cannot spontaneously bind to the (DETC)2-Fe2+ spin trap. Also, an additional water molecule as a co-ligand to NO did not give a stable H2O-(DETC)2-Fe2+-NO complex due to cleavage of Fe···OH2 bond on the DFT geometry optimization. We added these results to the manuscript (lines 142-151). 

Comment 4: Since DFT calculations were performed, what are the g values obtained from DFT, and how do they compare to the experimental ones? 

Response 4: We have estimated the g value by the DFT method and compared it with the experimental data (lines 137-138).

Reviewer 3 Report

Comments and Suggestions for Authors

The article under the title “Evaluation of Nitric Oxide-Donating Properties of 11H-in-deno[1,2-b]quinoxalin-11-one Oxime (IQ-1) by Electron Paramagnetic Resonance Spectroscopy” by Andrianov  and coworkers presents the experimental and theoretical results on the interaction between NO radical and spin trap, as the method for determination of the donating properties of IQ-1. The experimental section is well-described and the results are compared to the literature data. The article could be of potential interest to the readers of the Molecules, although there are some points that should be addressed before the final decision. Therefore, my recommendation is MAJOR REVISION.

The authors should answer the following:

1.       The authors should write NO as a radical species NO∙

2.        The authors should mention if the low temperature has some effect on the production of iron(II) complex or the production of NO radical

3.       The authors should clarify the source of naturally produced NO and add if other radicals could potentially react with the spin-trap

4.       Were the intensities dependent on the amount of added IQ-1?

5.       The authors have shown that the dissociation reaction was not thermodynamically spontaneous which can be expected due to the strong interaction between compounds. But it should be hypothesized if the reactions with some other compounds in the samples could be thermodynamically more spontaneous leading to the dissociation of the complex.

6.       The solvent effect should be explained based on the results in Table 1

7.       The applicability of the chosen level of theory should be verified by citing literature in which similar compounds were examined.  

Author Response

Comment 1: The authors should write NO as a radical species NO. 

Response 1: We understand. Nitric oxide is usually denoted as NO in numerous papers published in specialized journals (see, for example, https://www.sciencedirect.com/science/article/pii/S2213231720301828, https://doi.org/10.3390/molecules19079773, https://doi.org/10.3390/molecules24132470). This can be justified by the fact that nitric oxide is a relatively stable neutral molecule (not a radical intermediate). For example, the oxygen molecule is conventionally written as O2, while this is a diradical. We decided to write “NO” throughout the text of the manuscript and explicitly denoted the radical species just in Scheme 2 where the reaction mechanism is presented. We would like to leave it for the Journal editorial decision in accounting with Journal rules.

Comment 2: The authors should mention if the low temperature has some effect on the production of iron(II) complex or the production of NO radical.

Response 2: The formation of the iron(II) complex and the formation of the NO radical occur in vivo at physiological temperature. We used liquid nitrogen to freeze samples (liver or blood) to store samples before EPR measurements. At the temperature of liquid nitrogen, the complex does not decompose for a long time. On the other hand, low temperature is necessary for the registration of high-quality EPR spectra.

Comment 3: The authors should clarify the source of naturally produced NO and add if other radicals could potentially react with the spin-trap.

Response 3: The spin trap may interact with other radicals if they have the appropriate energy levels for binding. However, in the EPR spectrum, we observed only one complex ((DETC)2-Fe2+-NO), which gives a triplet signal with a certain g-factor. The others are either not visible by this method, or their signals have different shapes. We added the clarifying text in the Introduction (lines 42-45).

Comment 4: Were the intensities dependent on the amount of added IQ-1?

Response 4: We did not study the IQ-1 dose dependence of the signal. A single dose of the compound was used for the first study. We are planning an in-depth study of NO production depending on the dose of IQ-1 in our following work. We added this study limitation to the discussion part of the manuscript (lines 95-98).

Comment 5: The authors have shown that the dissociation reaction was not thermodynamically spontaneous which can be expected due to the strong interaction between compounds. But it should be hypothesized if the reactions with some other compounds in the samples could be thermodynamically more spontaneous leading to the dissociation of the complex.

Response 5: We added a discussion of these possibilities in the manuscript (lines 150-151).

Comment 6: The solvent effect should be explained based on the results in Table 1.

Response 6: We added the explanation of differences between the DFT results obtained in the gas phase and with the CPCM solvation model (lines 207-209).

Comment 7: The applicability of the chosen level of theory should be verified by citing literature in which similar compounds were examined.

Response 7: The choice of DFT functionals and basis sets is now described in the methodological part of the manuscript with the necessary literature references.

Round 2

Reviewer 1 Report

Comments and Suggestions for Authors

In my previous comments I asked the authors why the intensity of the  EPR signal from Hb-NO at addition of DETC + Fe2++IQ-1 is sharply increased? I did not receive any answer/

So, The paper can be issued if the authors will answer the question or notice in the text or they add to the text that they can not explain the experimental fact!

)

Author Response

Comment: In my previous comments I asked the authors why the intensity of the EPR signal from Hb-NO at addition of DETC + Fe2++IQ-1 is sharply increased? I did not receive any answer. So, The paper can be issued if the authors will answer the question or notice in the text or they add to the text that they can not explain the experimental fact!

Response: 

We appreciate the reviewer’s time and effort in reviewing this manuscript.

Components of the spin trap were administered to rats separately, including sodium DETC and a mixture of ferrous sulfate and sodium citrate. As a result, the organism must contain both insoluble (DETC)2-Fe2+ (the spin trap itself) and soluble forms of Fe2+, forming different complexes with NO, including (DETC)2-Fe2+-NO and Fe2+–NO species (see lines 131-135, 226-234). To make the text of the manuscript clearer to a reader, we pointed out that the introduction of spin trap components dramatically increases the amount of free Fe2+ species in the organism (line 125). Based on the DFT calculations, we showed that NO cannot be transferred to hemoglobin from its complex formed with the spin trap. Thus, the Gibbs free energy ΔGËš298 of the dissociation process estimated by the DFT method is equal to +25.0 kcal/mol, indicating very tight NO binding with the (DETC)2–Fe2+ spin trap (see lines 131-135). On the other hand, the dissociation constants of complexes formed with NO by different non-heme free Fe2+ species, including hydrated iron [Fe(H2O)6]2+, are within 10-2-10-6 M, while binding NO to HbFe2+ in R- and T-states is characterized by the dissociation constants of about 10-12 and 10-10 M, respectively [Cooper, C.E., Nitric oxide and iron proteins. Biochimica Et Biophysica Acta-Bioenergetics, 1999. 1411, 290-309]. Hence, NO can be captured by HbFe2+ from the free (non-spin-trap) Fe2+–NO species, giving the increase of HbFe2+–NO signals in the experimental EPR spectra (see lines 127-131). This mechanism may explain, why, as the Reviewer pointed out, the intensity of the ESR signal from Hb-NO increases sharply when DETC + Fe2+ are introduced. We noticed that signal levels of R- and T-conformers with IQ-1 were ~4-6 times higher compared to control and samples with the trap alone indicating that NO formed in vivo by oxidation of IQ-1 (see lines 114-115 and 280-281).

Reviewer 2 Report

Comments and Suggestions for Authors

The authors have answered all the questions I have. I can be published in the current form.

Author Response

Comment: The authors have answered all the questions I have. I can be published in the current form.

Response: Thank you very much for reviewing the manuscript.

Reviewer 3 Report

Comments and Suggestions for Authors

The authors have answered properly to the questions. The article is suitable for publication.

Author Response

Comment: The authors have answered properly to the questions. The article is suitable for publication.

Response: Thank you very much for reviewing the manuscript.